

# Sequencing 16S rRNA gene fragments using the PacBio SMRT DNA sequencing system

Patrick D. Schloss[1], Matthew L. Jenior[1], Charles C. Koumpouras[1], Sarah L. Westcott[1] and Sarah K. Highlander[2]

[1] Department of Microbiology and Immunology, University of Michigan, Ann Arbor, MI, USA
[2] Department of Genomic Medicine, J. Craig Venter Institute, La Jolla, CA, USA

## ABSTRACT

Over the past 10 years, microbial ecologists have largely abandoned sequencing 16S rRNA genes by the Sanger sequencing method and have instead adopted highly parallelized sequencing platforms. These new platforms, such as 454 and Illumina's MiSeq, have allowed researchers to obtain millions of high quality but short sequences. The result of the added sequencing depth has been significant improvements in experimental design. The tradeoff has been the decline in the number of full-length reference sequences that are deposited into databases. To overcome this problem, we tested the ability of the PacBio Single Molecule, Real-Time (SMRT) DNA sequencing platform to generate sequence reads from the 16S rRNA gene. We generated sequencing data from the V4, V3–V5, V1–V3, V1–V5, V1–V6, and V1–V9 variable regions from within the 16S rRNA gene using DNA from a synthetic mock community and natural samples collected from human feces, mouse feces, and soil. The mock community allowed us to assess the actual sequencing error rate and how that error rate changed when different curation methods were applied. We developed a simple method based on sequence characteristics and quality scores to reduce the observed error rate for the V1–V9 region from 0.69 to 0.027%. This error rate is comparable to what has been observed for the shorter reads generated by 454 and Illumina's MiSeq sequencing platforms. Although the per base sequencing cost is still significantly more than that of MiSeq, the prospect of supplementing reference databases with full-length sequences from organisms below the limit of detection from the Sanger approach is exciting.

# INTRODUCTION

Advances in sequencing technologies over the past 10 years have introduced considerable advances to the field of microbial ecology. Clone-based Sanger sequencing of the 16S rRNA gene has largely been replaced by various platforms produced by 454/Roche (e.g., *Sogin et al. (2006)*), Illumina (e.g., *Gloor et al. (2010)*), and IonTorrent (e.g., *Jünemann et al. (2012)*). It was once common to sequence fewer than 100 16S rRNA gene sequences from several samples using the Sanger approach (e.g., *McCaig et al. (1999)*). Now it is common

Corresponding author
Patrick D. Schloss,
pschloss@umich.edu

to generate thousands of sequences from each of several hundred samples (*Human Microbiome Consortium, 2012*). The advance in throughput has come at the cost of read length. Sanger sequencing regularly generated 800 nt per read and because the DNA was cloned, it was possible to obtain multiple reads per fragment to yield a full-length sequence from a representative single molecule. At approximately $8 (US) per sequencing read, most researchers have effectively decided that full-length sequences are not worth the increased cost relative to the cost of more recently developed approaches. There is still a clear need to generate high-throughput full-length sequence reads that are of sufficient quality that they can be used as references for analyses based on obtaining short sequence reads.

Historically, all sequencing platforms were created to primarily perform genome sequencing. When sequencing a genome, it is assumed that the same base of DNA will be sequenced multiple times and the consensus of multiple sequence reads is used to generate contigs. Thus, an individual base call may have a high error rate, but the consensus sequence will have a low error rate. To sequence the 16S rRNA gene researchers use conserved primers to amplify a sub-region from within the gene that is isolated from many organisms. Because the fragments are not cloned, it is not possible to obtain high sequence coverage from the same DNA molecule using these platforms. To reduce sequencing error rates it has become imperative to develop stringent sequence curation and denoising algorithms (*Schloss, Gevers & Westcott, 2011*; *Kozich et al., 2013*). There has been a tradeoff between read length, number of reads per sample, and the error rate. For instance, we recently demonstrated that using the Illumina MiSeq and the 454 Titanium platforms the raw error rate varies between 1 and 2% ( *Schloss, Gevers & Westcott, 2011*; *Kozich et al., 2013*). Yet, it was possible to obtain error rates below 0.02% by adopting various denoising algorithms; however, the resulting fragments were only 250-nt long. In the case of 454 Titanium, extending the length of the fragment introduces length-based errors and in the case of the Illumina MiSeq, increasing the length of the fragment reduces the overlap between the read pairs reducing the ability of each read to mutually reduce the sequencing error. Inadequate denoising of sequencing reads can have many negative effects including limited ability to identify chimeras (*Haas et al., 2011*; *Edgar et al., 2011*) and inflation of alpha- and beta-diversity metrics (*Kunin et al., 2010*; *Huse et al., 2010*; *Schloss, Gevers & Westcott, 2011*; *Kozich et al., 2013*). Illumina's MiSeq plaform enjoys widespread use in the field because of the ability to sequence 15–20 million fragments that can be distributed across hundreds of samples for less than $5,000 (US).

As these sequencing platforms have grown in popularity, there has been a decline in the number of full-length 16S rRNA genes being deposited into GenBank that could serve as references for sequence classification, phylogenetic analyses, and primer and probe design. This is particularly frustrating since the technologies have significantly improved our ability to detect and identify novel populations for which we lack full-length reference sequences. A related problem is the perceived limitation that the short reads generated by the 454 and Illumina platforms cannot be reliably classified to the genus or species level. Previous investigators have utilized simulations to demonstrate that increased read lengths usually increase the accuracy and sensitivity of classification against reference databases (*Wang et al., 2007*; *Liu et al., 2008*; *Werner et al., 2011*). There is clearly a need to develop sequencing

technologies that will allow researchers to generate high-quality, full-length 16S rRNA gene sequences in a high throughput manner.

New advances in single molecule sequencing technologies are being developed to address this problem. One approach uses a random barcoding strategy to fragment, sequence, and assemble full-length amplicons using Illumina's HiSeq platform (*Miller et al., 2013*; *Burke & Darling, 2014*). Although the algorithms appear to have minimized the risk of assembly chimeras, it is unclear what the sequencing error rate is by this approach. An alternative is the use of single molecule technologies that offer read lengths that are thousands of bases long. Although the Oxford Nanopore Technology has been used to sequence 16S rRNA genes (*Benítez-Páez, Portune & Sanz, 2016*), the platform produced by Pacific Biosciences (PacBio) has received wider attention for this application (*Fichot & Norman, 2013*; *Mosher et al., 2013*; *Mosher et al., 2014*; *Schloss et al., 2015*; *Singer et al., 2016*). The PacBio Single Molecule, Real-Time (SMRT) DNA Sequencing System ligates hairpin adapters (i.e., SMRTbells) to the ends of double-stranded DNA. Although the DNA molecule is linear, the adapters effectively circularize the DNA allowing the sequencing polymerase to process around the molecule multiple times (*Au et al., 2012*). According to Pacific Biosciences the platform is able to generate median read lengths longer than 8 kb with the P6-C4 chemistry; however, the single pass error rate is approximately 15%. Given the circular nature of the DNA fragment, the full read length can be used to cover the DNA fragment multiple times resulting in a reduced error rate. Therefore, one should be able to obtain multiple coverage of the full 16S rRNA gene at a reduced error rate.

Despite the opportunity to potentially generate high-quality, full-length sequences, it is surprising that the Pacific Biosciences platform has not been more widely adopted for sequencing 16S rRNA genes. Previous studies utilizing the technology have removed reads with mismatched primers and barcodes, ambiguous base calls, and low-quality scores (*Fichot & Norman, 2013*) or screened sequences based on the predicted error rate (*Singer et al., 2016*). Others have utilized the platform without describing the bioinformatic pipeline that was utilized (*Mosher et al., 2013*; *Mosher et al., 2014*). The only study to report the error rate of the platform for sequencing 16S rRNA genes with a mock community used the P4-C2 chemistry and obtained an error rate of 0.32% (*Schloss et al., 2015*), which is 16-fold higher than has been observed using the MiSeq or 454 platforms (*Schloss, Gevers & Westcott, 2011*; *Kozich et al., 2013*). In the current study, we assessed the quality of data generated by the PacBio sequencer using the improved P6-C4 chemistry and on-sequencer data processing. The goal was to determine whether this strategy could fill the need for generating high-quality, full-length sequence data on par with other platforms. We hypothesized that by modulating the 16S rRNA gene fragment length we could alter the read depth and obtain reads longer than are currently available by the 454 and Illumina platforms but with the same quality. To test this hypothesis, we developed a sequence curation pipeline that was optimized by reducing the sequencing error rate of a mock bacterial community with known composition. The resulting pipeline was then applied to 16S rRNA gene fragments that were isolated from soil and human and mouse feces.
**Table 1  Summary of the regions.** Summary of the primer pairs used to generate the 16S rRNA gene fragment fragments and the characteristics of each region.

| Region | Forward | Reverse | *E. coli* coordinates | Amplicon length | Sequences ($N$) |
|--------|---------|---------|-----------------------|-----------------|-----------------|
| V4 | GTGCCAGCMGCCGCGGTAA | GGACTACHVGGGTWTCTAAT | 515–806 | 253 | 21,934 |
| V1–V3 | AGRGTTTGATYMTGGCTCAG | ATTACCGCGGCTGCTGG | 8–534 | 490 | 36,545 |
| V3–V5 | CCTACGGGAGGCAGCAG | CCCGTCAATTCMTTTRAGT | 341–927 | 551 | 16,694 |
| V1–V5 | AGRGTTTGATYMTGGCTCAG | CCCGTCAATTCMTTTRAGT | 8–927 | 881 | 51,759 |
| V1–V6 | AGRGTTTGATYMTGGCTCAG | ACRACACGAGCTGACGAC | 8–1,078 | 1,033 | 64,599 |
| V1–V9 | AGRGTTTGATYMTGGCTCAG | GGYTACCTTGTTACGACTT | 8–1,510 | 1,464 | 61,721 |

## MATERIALS AND METHODS

### Community DNA

We utilized genomic DNA isolated from four communities. These same DNA extracts were previously used to develop an Illumina MiSeq-based sequencing strategy (*Kozich et al., 2013*). Briefly, we used a "Mock Community" composed of genomic DNA from 21 bacterial strains: *Acinetobacter baumannii* ATCC 17978, *Actinomyces odontolyticus* ATCC 17982, *Bacillus cereus* ATCC 10987, *Bacteroides vulgatus* ATCC 8482, *Clostridium beijerinckii* ATCC 51743, *Deinococcus radiodurans* ATCC 13939, *Enterococcus faecalis* ATCC 47077, *Escherichia coli* ATCC 70096, *Helicobacter pylori* ATCC 700392, *Lactobacillus gasseri* ATCC 33323, *Listeria monocytogenes* ATCC BAA-679, *Neisseria meningitidis* ATCC BAA-335, *Porphyromonas gingivalis* ATCC 33277, *Propionibacterium acnes* DSM 16379, *Pseudomonas aeruginosa* ATCC 47085, *Rhodobacter sphaeroides* ATCC 17023, *Staphylococcus aureus* ATCC BAA-1718, *Staphylococcus epidermidis* ATCC 12228, *Streptococcus agalactiae* ATCC BAA-611, *Streptococcus mutans* ATCC 700610, *Streptococcus pneumoniae* ATCC BAA-334. The mock community DNA is available through BEI resources (v3.1, HM-278D). Genomic DNAs from the three other communities were obtained using the MO BIO PowerSoil DNA extraction kit. The human and mouse fecal samples were obtained using protocols that were reviewed and approved by the University Committee on Use and Care of Animals (Protocol #PRO00004877) and the Institutional Review Board at the University of Michigan (Protocol #HUM00057066). The human stool donor provided informed consent.

### Library generation and sequencing

The DNAs were each amplified in triplicate using barcoded primers targeting the V4, V1–V3, V3–V5, V1–V5, V1–V6, and V1–V9 variable regions (Table 1). The primers were synthesized so that the 5′ end of the forward and reverse primers were each tagged with paired 16-nt symmetric barcodes (https://github.com/PacificBiosciences/Bioinformatics-Training/wiki/Barcoding-with-SMRT-Analysis-2.3) to allow multiplexing of samples within a single sequencing run. Methods describing PCR, amplicon cleanup, and pooling were described previously (*Kozich et al., 2013*). The SMRTbell adapters were ligated onto the PCR products and the libraries were sequenced by Pacific Biosciences using the P6-C4 chemistry on a PacBio RS II SMRT DNA Sequencing System. Diffusion Loading was used for regions V4, V1–V3, and V3–V5 and MagBead loading was used for regions V1–V5, V1–V6, and V1–V9. Each region was sequenced separately using movies ranging

in length between 180 and 360 min. The sequences were processed using pbccs (v.3.0.1; https://github.com/PacificBiosciences/pbccs), which generates predicted error rates using a proprietary algorithm.

## Data analysis

All sequencing data were curated using mothur (v1.36) (*Schloss et al., 2009*) and analyzed using the R programming language (*R Core Team, 2016*). The raw data can be obtained from the Sequence Read Archive at NCBI under accession SRP051686, which are associated with BioProject PRJNA271568. This accession and bioproject also contain data from the same samples sequenced using the P4-C2 chemistry. Several specific features were incorporated into mothur to facilitate the analysis of PacBio sequence data. First, because non-ambiguous base calls are assigned to Phred quality scores of zero, the consensus fastq files were parsed so that scores of zero were interpreted as corresponding to an ambiguous base call (i.e., N) in the fastq.info command using the pacbio=T option. Second, because the consensus sequence can be generated in the forward and reverse complement orientations, a checkorient option was added to the trim.seqs command in order to identify the proper orientation. These features were incorporated into mothur v.1.30. Because chimeric molecules can be generated during PCR and would artificially inflate the sequencing error, it was necessary to remove these data prior to assessing the error rate. Because we knew the true sequences for the strains in the mock community we could calculate all possible chimeras between strains in the mock community (*in silico* chimeras). If a sequence read was 3 or more nucleotides more similar to an *in silico* chimera than it was to a non-chimeric reference sequence, it was classified as a chimera and removed from further consideration. Identification of *in silico* chimeras and calculation of sequencing error rates was performed using the seq.error command in mothur (*Schloss, Gevers & Westcott, 2011*). *De novo* chimera detection was also performed on the mock and other sequence data using the abundance-based algorithm implemented in UCHIME (*Edgar et al., 2011*). Sequences sequences were aligned against a SILVA-based reference alignment (*Pruesse et al., 2007*) using a profile-based aligner (*Schloss, 2009*) and were classified against the SILVA (v123) (*Pruesse et al., 2007*), RDP (v10) (*Cole et al., 2013*), and greengenes (v13_8_99) (*Werner et al., 2011*) reference taxonomies using a negative Bayesian classifier implemented within mothur (*Wang et al., 2007*). Sequences were assigned to operational taxonomic units using the average neighbor clustering algorithm with a 3% distance threshold ( *Schloss & Westcott, 2011* ). Detailed methods including this paper as an R markdown file are available as a public online repository (http://github.com/SchlossLab/Schloss_PacBio16S_PeerJ_2016).

## RESULTS AND DISCUSSION

### The PacBio error profile and a basic sequence curation procedure

To build a sequence curation pipeline, we first needed to characterize the error rate associated with sequencing the 16S rRNA gene. Using the consensus sequence obtained from at least 3 sequencing passes of each fragment, we observed an average sequencing error rate of 0.65%. Insertions, deletions, and substitutions accounted for 31.2, 17.9, and 50.9% of the errors, respectively. The substitution errors were equally likely and all four
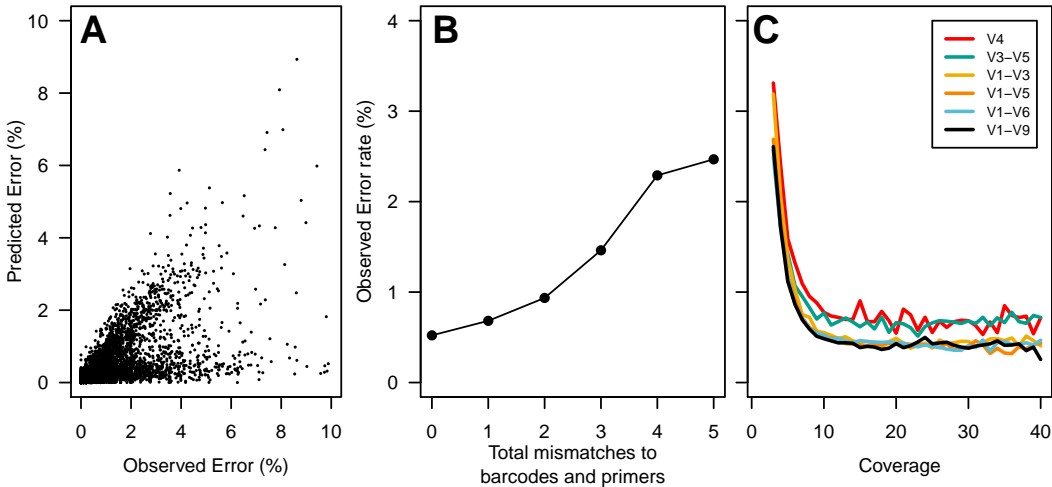

**Figure 1** **Summary of errors in data generated using PacBio sequencing platform to sequence various regions within the 16S rRNA gene.** The predicted error rate using PacBio's sequence analysis algorithm correlated well with the observed error rate (Pearson's $R$ : $-0.67$; (A). Because of the large number of sequences, we randomly selected 5% of the data to show in (A). The sequencing error rate of the amplified gene fragments increased with mismatches to the barcodes and primers (B). The sequencing error rate declined with increased sequencing coverage; however, increasing the sequencing depth beyond 10-fold coverage had no meaningful effect on the sequencing error rate (C). The scale of they $y$-axis in B and C are the same.

bases were equally likely to cause insertion errors. Interestingly, guanines (39.4%) and adenines (24.3%) were more likely to be deleted than cytosines (18.3%) or thymidines (18.0%). The PacBio quality values varied between 2 and 93. Surprisingly, the percentage of base calls that had the maximum quality value did not vary among correct base calls (80.5%), substitutions (80.0%), and insertions (80.4%). It was not possible to use the individual base quality scores to screen sequence quality as has been possible in past studies using the Phred-based quality scores that accompany data generated using the 454 and Illumina technologies. We did observe a strong correlation between our observed error rate and the predicted error rate as calculated by the PacBio software (Pearson's R: $-0.67$; Fig. 1A).

We established a simple curation procedure by culling any sequence that had a string of the same base repeated more than eight times or did not start and end at the expected alignment coordinates for that region of the 16S rRNA gene. This reduced the experiment-wide error rate from 0.68 to 0.65%. This basic procedure resulted in the removal of between 0.714 (V1–V3) and 9.47 (V1–V9)% of the reads (Table 2). Although the percentage of reads removed increased with the length of the fragment, there was no obvious relationship between fragment length and error rate (Fig. 2).

## Identifying correlates of increased sequencing error

In contrast to the 454 and Illumina-based platforms where the sequencing quality decays with length, the consensus sequencing approach employed by the PacBio sequencer is thought to generate a uniform distribution of errors. This makes it impossible to simply trim sequences to high-quality regions. Therefore, we sought to identify characteristics

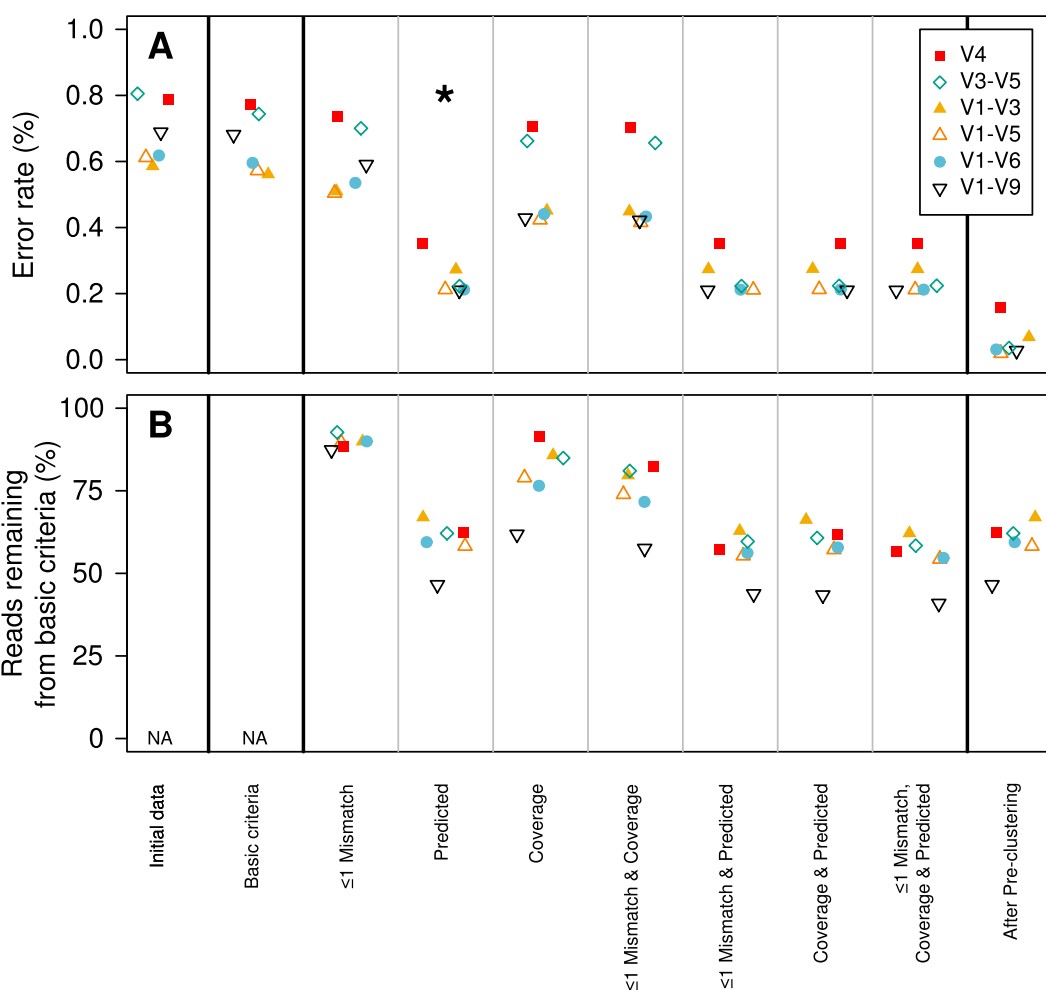

**Figure 2  Change in error rate (A) and the percentage of sequences that were retained (B) when using various sequence curation methods.** The condition that was used for downstream analyses is indicated by the star. The plotted numbers represent the region that was sequenced. For example ''19'' represents the data for the V1–V9 region.

**Table 2  Summary of error rates.** Summary of the error rates and number of observed OTUs for each region.

| Region | Error rate following (%) | | | Reads remaining (%) | Average no. of OTUs[a] | | | | |
|---|---|---|---|---|---|---|---|---|---|
| | Basic | Predicted | Precluster | | Mock[b] | Mock[c] | Soil | Mouse | Human |
| V4 | 0.77 | 0.35 | 0.158 | 62.58 | 33.4 | 48.9 | 629.4 | 212.6 | 116.9 |
| V1–V3 | 0.56 | 0.27 | 0.068 | 67.01 | 20.5 | 27.7 | 726.0 | 107.3 | 106.4 |
| V3–V5 | 0.74 | 0.22 | 0.035 | 63.15 | 21.1 | 31.2 | NA | 112.3 | 83.8 |
| V1–V5 | 0.57 | 0.21 | 0.019 | 59.29 | 19.8 | 27.2 | 694.4 | 65.9 | 84.9 |
| V1–V6 | 0.60 | 0.21 | 0.031 | 59.90 | 20.0 | 46.5 | 693.0 | 75.8 | 89.4 |
| V1–V9 | 0.68 | 0.21 | 0.027 | 51.33 | 19.5 | 40.4 | NA | 102.5 | 117.3 |

**Notes.**
[a]The number of OTUs is based on rarefaction of each sample to 1,000 sequences per sample; cells labeled ND reflect samples that did not have at least 1,000 sequences.
[b]Number of OTUs in the mock community when all chimeras were removed; in the absence of chimeras and sequencing errors, there should be 19 OTUs for all three regions.
[c]Number of OTUs in the mock community when chimeras were removed using UCHIME.

within sequences that would allow us to identify and remove those sequences with errors using three different approaches. First, we hypothesized that errors in the barcode and primer would be correlated with the error rate for the entire sequence. We observed a strong relationship between the number of mismatches to the barcodes and primers and the error rate of the rest of the sequence fragment (Fig. 1B). Although allowing no mismatches to the barcodes and primers yielded the lowest error rate, that stringent criterion removed a large fraction of the reads from the dataset. Allowing at most one mismatch only marginally increased the error rate while retaining more sequences in the dataset (Fig. 2). Second, we hypothesized that increased sequencing coverage should yield lower error rates. We found that once we had obtained 10-fold coverage of the fragments, the error rate did not change appreciably (Fig. 1C). When we compared the error rates of reads with at least 10-fold coverage to those with less coverage, we reduced the error rate by 8.48–37.08% (Fig. 2). Third, based on the observed correlation between the predicted and observed error rates, we sought to identify a minimum predicted error rate that would allow us to reduce the observed error rate. The average observed error rate for sequences with predicted error rates between 0.01 and 0.10% was linear. We decided to use a threshold of 0.01% because a large number of sequence reads were lost when we used a smaller threshold. When we used this threshold, we were able to reduce the error rate by 51.4–70.0% (Fig. 2). Finally, we quantified the effect of combining filters. We found that any combination of filters that included the predicted error rate threshold had the most significant impact on reducing the observed error rate. Furthermore, the inclusion of the mismatch and coverage filters had a negligible impact on error rates, but had a significant impact on the number of sequences included in the analysis. For instance, among the V1–V9 data, requiring sequences to have a predicted error rate less than 0.01% resulted in a 69.2% reduction in error and resulted in the removal of 53.5% of the sequences. Adding the mismatch or coverage filter had no effect on the reduction of error, but resulted in the removal of 56.2 and 56.6% of the sequences, respectively. Use of all filters had no impact on the reduction in the observed error rate, but resulted in the removal of 59.1% of the sequences. The remainder of this paper only uses sequences with a predicted error rate less than 0.01%.

## Pre-clustering sequences to further reduce sequencing noise

Previously, we implemented a pre-clustering algorithm where sequences were sorted by their abundance in decreasing order and rare sequences are clustered with a more abundant sequence if the rare sequences have fewer mismatches than a defined threshold when compared to the more abundant sequence (*Huse et al., 2010*; *Schloss, Gevers & Westcott, 2011*). The recommended threshold was a 1-nt difference per 100-nt of sequence data. For example, the threshold for 250 bp fragment from the V4 region would be 2 nt or 14 for the 1,458 bp V1–V9 fragments. This approach removes residual PCR and sequencing errors while not overwhelming the resolution needed to identify OTUs that are based on a 3% distance threshold. The tradeoff of this approach is that one would be unable to differentiate V1–V9 sequences that truly differed by less than 14 nt. When we applied this approach to our PacBio data, we observed a reduction in the error rate between 33.0 (V1–V3) and 48.7% (V1–V9). The final error rates varied between 0.02 (V1–V5)

and 0.2% (V4). The full-length (i.e., V1–V9) fragments had an error rate of 0.03% (Fig. 2; Table 2), this is similar to what we have previously observed using the 454 and Illumina MiSeq platforms (0.02%) (*Schloss, Gevers & Westcott, 2011*; *Kozich et al., 2013*).

## Effects of error rates on OTU assignments

The sequencing error rate is known to affect the number of OTUs that are observed (*Schloss, Gevers & Westcott, 2011*). For each region, we determined that if there were no chimeras or PCR or sequencing errors, then we would expect to find 19 OTUs. When we achieved perfect chimera removal, but allowed for PCR and sequencing errors, we observed between 0.5 (V1–V9) and 14.4 (V4) extra OTUs (Table 2). The range in the number of extra OTUs was largely explained by the sequencing error rate (Pearson's $R = 1.0$). Next, we determined the number of OTUs that were observed when we used UCHIME to identify chimeric sequences. Under these more realistic conditions, we observed between 8.2 (V1–V5) and 29.9 (V4) extra OTUs. Finally, we calculated the number of OTUs in the soil, mouse, and human samples using the same pipeline with chimera detection and removal based on the UCHIME algorithm. Surprisingly, there was not a clear relationship across sample type and region. Again, we found that there was a strong correlation between the number of observed OTUs and the error rate for the mouse ($R = 0.95$) and human samples ($R = 0.60$). These results underscore the effect of sequencing error on the inflation of the number of observed OTUs.

## Classification varies by region, environment, and database

We classified all of the sequence data we generated using the naïve Bayesian classifier using the RDP, SILVA, and greengenes reference taxonomies (Fig. 3). In general, increasing the length of the region improved the ability to assign the sequence to a genus or species. Interestingly, each of the samples we analyzed varied in the ability to assign its sequences to the depth of genus or species. Furthermore, the reference database that did the best job of classifying the sequences varied by sample type. For example, the SILVA reference did the best for the human feces and soil samples and the RDP did the best for the mouse feces samples. An advantage of the greengenes database is that it contains information for 2,514 species-level lineages for 11% of the reference sequences; the other databases only provided taxonomic data to the genus level. There was a modest association between the length of the fragment and the ability to classify sequences to the species-level for the human samples; there was no such association for the mouse and soil samples. In fact, at most 6.2% of the soil sequences and 4.3% of the mouse sequences could be classified to a species. These results indicate that the ability to classify sequences to the genus or species level is a function of read length, sample type, and the reference database.

## Sequencing errors are not random

Above, we described that although there was no obvious bias in the substitution or insertion rate, we did observe that guanines and adenines were more likely to be deleted than cytosines or thymidines. This lack of randomness in the error profile suggested that there might be a systematic non-random distribution of the errors across the sequences. This would manifest itself by the creation of duplicate sequences with the same error. We identified all

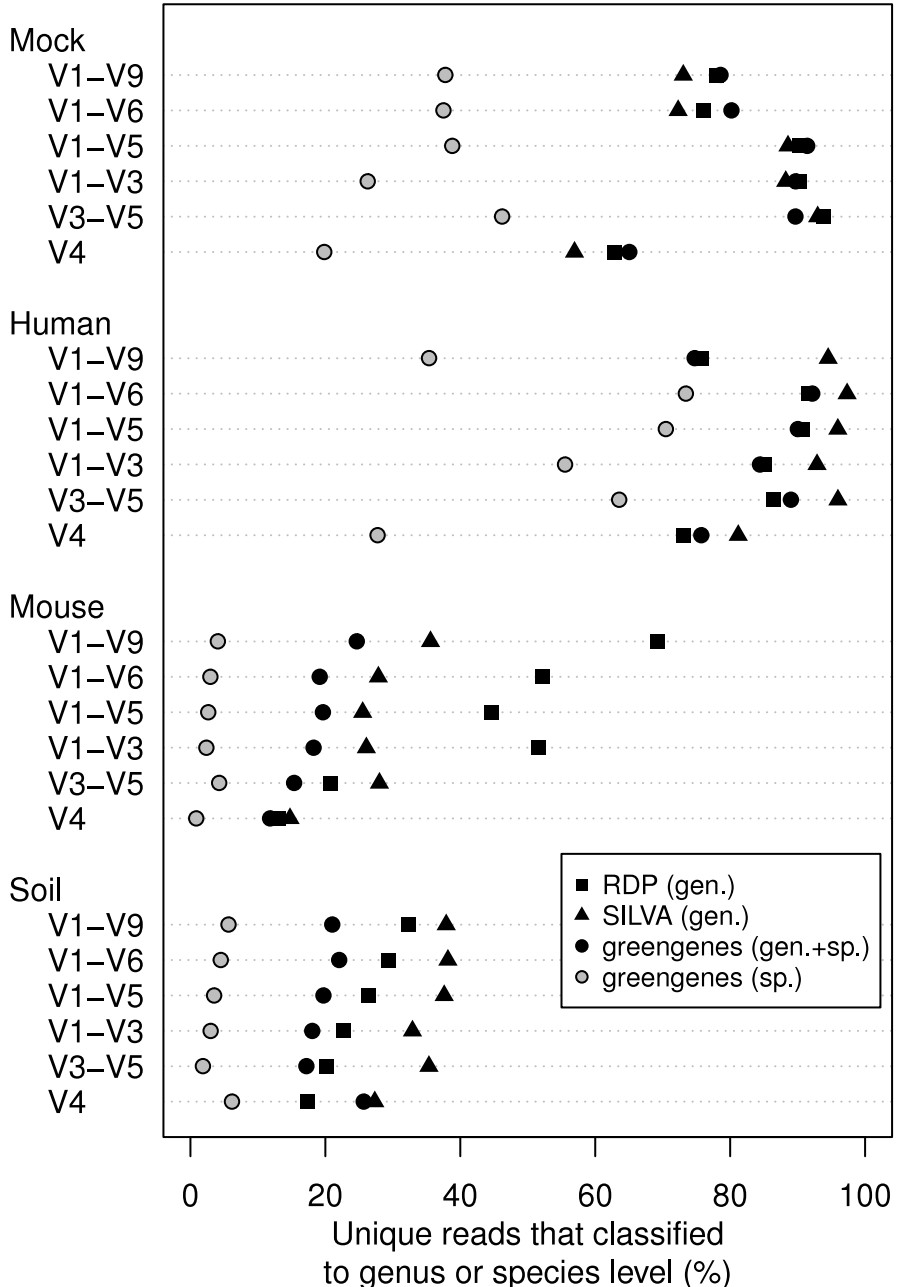

**Figure 3** **Percentage of unique sequences that could be classified.** Classifications were performed using taxonomy references curated from the RDP, SILVA, or greengenes databases for the four types of samples that were sequenced across the six regions from the 16S rRNA gene. Only the greengenes taxonomy reference provided species-level information.

of the mock community sequences that had a 1-nt difference to the true sequence (Fig. 4). For these three regions, between 70.7 and 88.9 of the sequences with 1-nt errors were only observed once. We found that the frequency of the most abundant 1-nt error paralleled the number of sequences. Surprisingly, the same 1-nt error appeared 1,954 times (0.02%) in

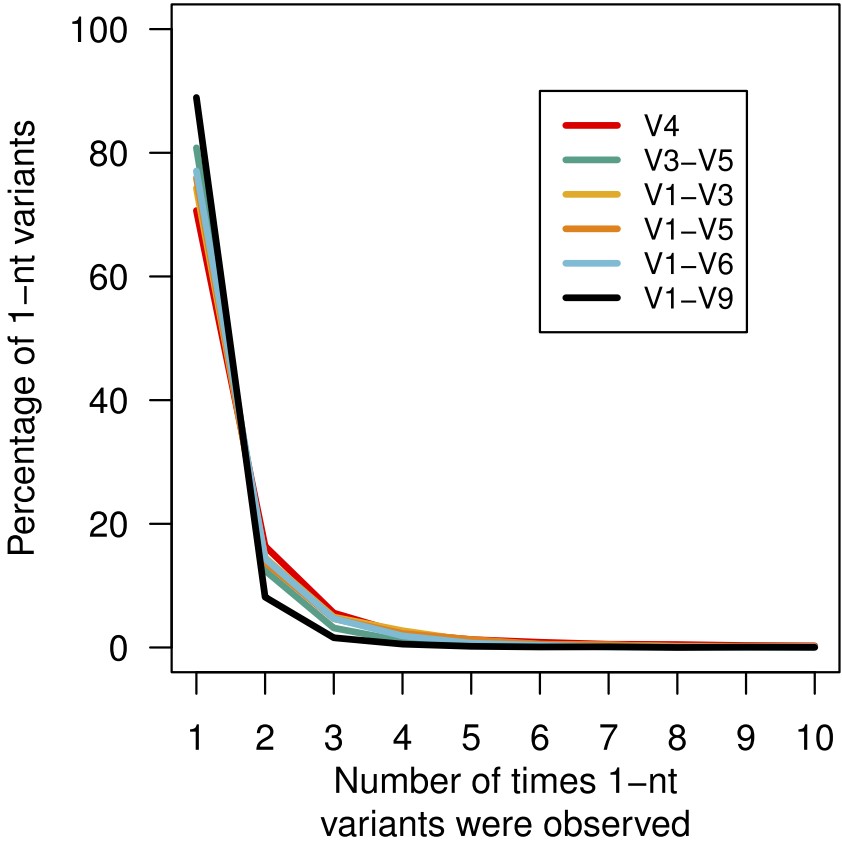

**Figure 4  Percentage of 1-nt variants that occurred up to ten times.** Sequences that were 1 nt different from the mock community reference sequences were counted to determine the number of times each variant appeared by region within the 16S rRNA gene.

the V1–V6 mock data and another 1-nt error appeared 1,070 times (0.03%) in the V1–V9 mock data. Contrary to previous reports (*Carneiro et al., 2012*; *Koren et al., 2012*), these results indicate that reproducible errors occur with the PacBio sequencing platform and that they can be quite abundant. Through the use of the pre-clustering step described above these 1-nt errors would be ameliorated; however, this result indicates that caution should be used when attempting to use fine-scale OTU definitions.

## CONCLUSIONS

The various sequencing platforms that are available to microbial ecologists are able to fill unique needs and have their own strengths and weaknesses. For sequencing the 16S rRNA gene, the 454 platform is able to generate a moderate number of high-quality 500-nt sequence fragments (error rates below 0.02%) (*Schloss, Gevers & Westcott, 2011*) and the MiSeq platform is able to generate a large number of high-quality 250-nt sequence fragments (error rates below 0.02%) (*Kozich et al., 2013*). The promise of the PacBio sequencing platform was the generation of high-quality near full-length sequence fragments. As we have shown in this study, it is possible to generate near full-length

sequences with error rates that are slightly higher, but comparable to the other platforms (i.e., 0.03%). With the exception of the V4 region (0.2%), the error rates were less than 0.07%. When we considered the shorter V4 region, which is similar in length to what is sequenced by the MiSeq platform, the error rates we observed with the PacBio platform were nearly 8-fold higher than what has previously been reported on the other platforms. It was unclear why these shorter reads had such a high error rate relative to the other regions. At this point, the primary limitation of generating full-length sequences on the PacBio platform is the cost of generating the data and accessibility to the sequencers.

The widespread adoption of the 454 and MiSeq platforms and decrease in the use of Sanger sequencing for the 16S rRNA gene has resulted in a decrease in the generation of the full-length reference sequences that are needed for performing phylogenetic analyses and designing lineage specific PCR primers and fluorescent *in situ* hybridization (FISH) probes. It remains to be determined whether the error rates we observed for full-length sequences are prohibitive for these applications. We can estimate the distribution of errors assuming that the errors follow a binomial distribution along the length of the 1,500 nt gene with the error rate that we achieved from the V1–V9 mock community data prior to pre-clustering the sequences, which was 0.2%. Under these conditions one would expect 4.3% of the sequences to have no errors and 50% of the sequences would have at least 3 errors. After applying the pre-clustering denoising step, the error rate drops by 7.7-fold to 0.03%. With this error rate, we would expect 66.3% of the sequences to have no sequencing errors. The cost of the reduced error rate is the loss of resolution among closely related sequences.

Full-length sequences are frequently seen as a panacea to overcome the limitations of taxonomic classifications. The ability to classify each of our sample types benefited from the generation of full-length sequences. It was interesting that the benefit varied by sample type and database. For example, using the mouse libraries, the ability to classify each of the regions differed by less than 5% when classifying against the SILVA and greengenes databases. The effect of the database that was used was also interesting. The RDP database outperformed the other databases for the mouse samples and the SILVA database outperformed the others for the human and soil samples. The three databases were equally effective for classifying the mock community. Finally, since only the greengenes database provided species-level information for its reference sequences it was the only database that allowed for resolution of species-level classification. The sequences from the mouse and soil libraries were not effectively classified to the species level (all less than 10%). In contrast, classification of the human libraries resulted in more than 40% of the sequences being classified to a genus, regardless of the region. These data demonstrate that for the samples we analyzed, the length of the sequence fragment was not as significant a factor in classification as the choice of database.

The development of newer sequencing technologies continue to advance and there is justifiable excitement to apply these technologies to sequence the 16S rRNA gene. Although it is clearly possible to generate sequencing data from these various platforms, it is critical that we assess the platforms for their ability to generate high-quality data and the particular niche that the new approach will fill. With this in mind, it is essential that researchers utilize mock communities as part of their experimental design so that they can quantify

their error rates. The ability to generate near full-length 16S rRNA gene sequences is an exciting advance that will hopefully expand our ability to improve the characterization of microbial communities.

## ACKNOWLEDGEMENTS

The Genomic DNA from Microbial Mock Community A (Even, Low Concentration, v3.1, HM-278D) was obtained through the NIH Biodefense and Emerging Infections Research Resources Repository, NIAID, NIH as part of the Human Microbiome Project.

### Funding

This study was supported by grants from the NIH (R01HG005975, R01GM099514 and P30DK034933 to PDS and U54HG004973 to SKH). The funders had no role in study design, data collection and analysis, decision to publish, or preparation of the manuscript.

### Grant Disclosures

The following grant information was disclosed by the authors:
NIH: R01HG005975, R01GM099514, P30DK034933, U54HG004973.

### Competing Interests

Sarah K. Highlander is an employee of JCVI.

### Author Contributions

- Patrick D. Schloss conceived and designed the experiments, performed the experiments, analyzed the data, contributed reagents/materials/analysis tools, wrote the paper, prepared figures and/or tables, reviewed drafts of the paper.
- Matthew L. Jenior and Charles C. Koumpouras performed the experiments, reviewed drafts of the paper.
- Sarah L. Westcott analyzed the data, contributed reagents/materials/analysis tools, reviewed drafts of the paper.
- Sarah K. Highlander conceived and designed the experiments, performed the experiments, contributed reagents/materials/analysis tools, reviewed drafts of the paper.

### Human Ethics

The following information was supplied relating to ethical approvals (i.e., approving body and any reference numbers):

The University of Michigan Institutional Review Board (HUM00057066).

### Animal Ethics

The following information was supplied relating to ethical approvals (i.e., approving body and any reference numbers):

University of Michigan Committee on Use and Care of Animals (PRO00004877).

## DNA Deposition

The following information was supplied regarding the deposition of DNA sequences:

The raw data can be obtained from the Sequence Read Archive at NCBI under accession SRP051686, which are associated with BioProject PRJNA271568.

## Data Availability

Detailed methods including this paper as an R markdown file are available as a public online repository (https://github.com/SchlossLab/Schloss_PacBio16S_PeerJ_2016).

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
