# Peer review of "Sequencing 16S rRNA gene fragments using the PacBio SMRT DNA sequencing system"

_PeerJ, doi:10.7717/peerj.1869_

## Round 0.1 · original submission · Major Revisions

Dear Dr. Schloss, Dr. Highlander and colleagues,

Thank you for submitting a manuscript that is of interest to a broad readership of scientists applying high-throughput NGS in particular with respect to microbial community analysis.

We have now obtained reviews from three expert referees from different areas of the field who underscore this general importance of your work.

While they share the opinion that the work is indeed very useful and timely, they have a few major concerns and questions, for instance regarding the bioinformatics pipeline and the impact of intermittent improvements to the PacBio chemistry which might improve 16S amplicon sequencing accuracy. Answers to all these reviewers’ questions will be crucial to clarify where exactly the determined higher error rate in comparison to the other platforms resulted from; and whether making changes to the PCR template preparation system, increasing read length and/or coverage, and changing bioinformatics filtering might be relevant to reducing the error rate of the platform. All the reviewers’ questions need to be addressed meticulously, since these aspects are very important for the comparison of platform performance with other currently available NGS/16S platforms and are quite certain to impact on their current and future use.

We are confident that you can fully and convincingly address the reviewers’ concerns and comments, in order to resubmit a revised version that will be very useful to the community.

Best regards,
Christine Josenhans

Reviewer 1 ·

Basic reporting

I am not a bioinformatician, so I cannot judge the appropriateness of the analytical pipeline. I trust that other reviewers will complement this limitation. The paper is very well written and I have no concerns with respect to Basic Reporting. Systematic and unbiased studies that carefully investigate the error rates of competing NGS technologies are important both for guiding scientists when they make technology choices and to evaluate published studies using those technologies. Even if the paper is a technical one, the analyses have very important implications for a large community of scientists, so that it clearly represents an appropriate "unit of publication".

Experimental design

Schloss et al. have used PacBio SMRT sequencing technology to sequence 16S rDNA, determined the error rates, compared them to published error rates of the widely used competing 454 and MiSeq platforms, and finally comment on the impact of differences of error rates on the results of microbiota composition analysis. The paper is clearly written and comes to the conclusion that PacBio technology generates too many errors and too little sequence to be competitive with 454 and MiSeq platforms.
While this all is presented in a convincing fashion, the study has one major limitation. The authors come to a sweeping conclusion about a technology using a version of the platform that is no longer available and has been superseded by two new generations of polymerase enzyme and sequencing chemistry. Since the paper focusses on a single technology, its conclusions using the P4-C2 chemistry are of very limited value to scientists currently working with the technology (current version: P6-C4). While it has to be appreciated that studies like these shoot at a moving target, the authors would at the very least have to make it very clear that their conclusions are only valid for the older version of the platform. The very strong statements of the current manuscript convey a message that ignores reported improvements.

Validity of the findings

Ideally, the authors should include additional analyses on the V1-V9 amplicon using the latest version of the PacBio platform. I believe this could be done without too much additional effort and greatly strengthen the impact of the paper. Alternatively, the authors need to place very visible disclaimers making it clear that all analyses are based on data generated with the P4-C2 chemistry, which is obsolete.
Error rates will also be influenced by the initial amplification. Were the exact same conditions used to generate the amplicons used for PacBio sequencing that were also used in the publication by Kozich et al., that serves as the reference for the error rates of the MiSeq platform.

Reviewer 2 ·

Basic reporting

No Comments

Experimental design

No Comments

Validity of the findings

No Comments

Additional comments

In this study the authors explored the possibility of utilizing PacBio, a platform that gives much longer reads than the current widely used sequencers available. The known higher error rate of the platform generally deems it not applicable to 16S profiling, thus, the authors developed analytical approaches to reduce the final error to enable better use of the data. This is a very useful study to the community, as it would clearly help researchers to decide on the feasibility of the platform.

A general comment is, given the longer reads possible with this platform, perhaps there might not be need to focus on only one or a few commonly used hyper-variable region, but to focus on the nearly-full-length sequences. Considering the 3% divergence threshold commonly used for species definition, on the order of 40 errors could be permitted for a sequence length of 1500, and the goal of accurate genus classification and OTU clustering might be adequate. This may of course have other consequences when most analyses are done de novo, but I would suggest reference based analysis for chimera removal and OTU clustering. Although the authors have already done a very fine job with their assessment, perhaps they could address this.

Specific comments:

Lines 140-141: Could the authors clarify the nature of the quality score from PacBio (Figure 1A does not appear to be comparable to that seen from other platforms)? How does it compare to that of Illumina/454 and how does mothur cope with the differences if there are any?


Lines 279-282: “We found that the frequency of the most abundant 1-nt error paralleled the
number of sequences. There were two sequences in the V4 dataset that occurred 76 times, one
sequence in the V1-V5 dataset that occurred 30 times, and one sequence in the V3-V5 dataset
that occurred 17 times.” Are there any general characteristics of the most abundant errors that could provide insight into why certain errors tend to occur? e.g. high AT content or high nucleotide diversity overall (or in a particular sliding window)?

Figure 2 is bit difficult to follow with this “numbering” scheme, suggest to change to colored lines.

·

Basic reporting

.

Experimental design

.

Validity of the findings

.

Additional comments

Summary: This is an important paper given the large interest in conducting 16S surveys, and the wide range of sequencing technologies available to researchers. The authors are well-versed in the technical issues associated with the use of this data, and have presented a compelling case that this platform may not offer sufficient advantages in terms of error rates (which are in fact higher) or phylogenetic resolution to compensate for the lower number of reads compared to other technologies. I strongly recommend publication after minor revision.

Comments: In my first few readings of the manuscript it was unclear how the additional sequencing depth was taken into account. In the beginning of the results it states the error rates were ~1.8% on average. But it was unclear whether this is the baseline error rate for a single pass, or whether it was the result of taking a consensus of multiple passes. Since the introduction states the single pass error is ~15%, it must be the latter which makes sense given the results, but I’d recommend explaining this at the beginning of the results section.

It would also be helpful to explain how the multiple reads are used to reduce error rates. For example, if the single pass error is 15%, then the chances of seeing an incorrect nucleotide on 5 or more out of 10 passes would be something like 0.006%, which seems pretty good. But I’m guessing that non-random errors invalidate this simple estimate, but I cant tell for sure because there’s no markings on the y-axis to figure 1C. If the high error rates stem from the fact that coverage rates are generally much lower than 10X, that would be good to know, because it means the platform could potentially be useful if read lengths are increased.

Finally, it was not clear if this study was based on a single sequencing run at a single facility. If so, it might be important to point out that results could vary from run to run, especially if read lengths/sequencing depth plays a big role.

Minor points:

• line 215: "whether" should be deleted.
• line 233: “one would unable…”
• line 243: should say "when we achieved...".
• lines 244ff.: in one place, they say they observed between 6 and 63.1 extra OTUs. in another place, between 7.4 and 86.8 extra OTUs. how do you have a fractional OTU?
• line 246: "sequence" should be plural.
• Figure 2 is illegible. why not just have colored points or something? or add more left-right jitter?
• line 280: if I understand correctly, it seems that some errors occurred once, but a small number occurred many times. That suggests the latter type may be errors during PCR or even cell division, which are not necessarily relevant to the sequencing methodology. If they are biases of the platform, however, that would be a major problem, so it would be good to have some clarification on this point.

---

## Round 0.2 · accepted · Accept

Dear Dr. Schloss,

I congratulate you on an interesting and timely study, which will be widely read (and already has been read, due to preprints). Thank you for thoroughly revising the study including new experiments and experimental design.

Best regards,
Christine Josenhans

Reviewer 1 ·

Basic reporting

No Comments

Experimental design

No Comments

Validity of the findings

No Comments

Additional comments

The authors are to be commended for having redone the complete experiment and analysis using the current PacBio sequencing technology. My concerns have been fully addressed and I recommend immediate publication.

Minor comment: There is a glitch in the sentence spanning lines 177 and 178 which could be corrected during production.